# A leaf reflectance-based crop yield modeling in Northwest Ethiopia

**Gizachew Ayalew Tiruneh** [1¤] *, **Derege Tsegaye Meshesha**[2], **Enyew Adgo** [2], **Atsushi Tsunekawa** [3], **Nigussie Haregeweyn**[4], **Ayele Almaw Fenta**[3], **José Miguel Reichert** [5]

**1** Faculty of Agriculture and Environmental Sciences, Debre Tabor University, Debre Tabor, Ethiopia, **2** College of Agriculture and Environmental Sciences, Bahir Dar University, Bahir Dar, Ethiopia, **3** Arid Land Research Center, Tottori University, Hamasaka, Tottori, Japan, **4** International Platform for Dryland Research and Education, Tottori University, Hamasaka, Tottori, Japan, **5** Soils Department, Universidade Federal de Santa Maria (UFSM), Santa Maria, RS, Brazil

¤ Current address: Department of Natural Resources Management, College of Agriculture and Environmental Sciences, Bahir Dar University, Bahir Dar, Ethiopia
* tiruneh1972@gmail.com

## Abstract

Crop yield prediction provides information to policymakers in the agricultural production system. This study used leaf reflectance from a spectroradiometer to model grain yield (GY) and aboveground biomass yield (ABY) of maize (*Zea mays* L.) at Aba Gerima catchment, Ethiopia. A FieldSpec IV (350–2,500 nm wavelengths) spectroradiometer was used to estimate the spectral reflectance of crop leaves during the grain-filling phase. The spectral vegetation indices, such as enhanced vegetation index (EVI), normalized difference VI (NDVI), green NDVI (GNDVI), soil adjusted VI, red NDVI, and simple ratio were deduced from the spectral reflectance. We used regression analyses to identify and predict GY and ABY at the catchment level. The coefficient of determination ($R^2$), the root mean square error (RMSE), and relative importance (RI) were used for evaluating model performance. The findings revealed that the best-fitting curve was obtained between GY and NDVI ($R^2$ = 0.70; RMSE = 0.065; P < 0.0001; RI = 0.19), followed by EVI ($R^2$ = 0.65; RMSE = 0.024; RI = 0.61; P < 0.0001). While the best-fitting curve was obtained between ABY and GNDVI ($R^2$ = 0.71; RI = 0.24; P < 0.0001), followed by NDVI ($R^2$ = 0.77; RI = 0.17; P < 0.0001). The highest GY (7.18 ton/ha) and ABY (18.71 ton/ha) of maize were recorded at a soil bunded plot on a gentle slope. Combined spectral indices were also employed to predict GY with $R^2$ (0.83) and RMSE (0.24) and ABY with $R^2$ (0.78) and RMSE (0.12). Thus, the maize's GY and ABY can be predicted with acceptable accuracy using spectral reflectance indices derived from spectroradiometer in an area like the Aba Gerima catchment. An estimation model of crop yields could help policy-makers in identifying yield-limiting factors and achieve decisive actions to get better crop yields and food security for Ethiopia.

## Introduction

Ethiopia has Agricultural Development Led Industrialization -based economy, employing 75% of the labor pool and sharing about 40% of the country's total economic output [1–3]. Rain-

**Data Availability Statement:** All relevant data are within the paper and its Supporting Information files.

**Funding:** The research was funded by the Science and Technology Research Partnership for Sustainable Development (grant number JPMJSA1601), Japan Science and Technology Agency/Japan International Cooperation Agency (JICA). Gizachew Ayalew received the fund award. The funders had no role in study design, data collection and analysis, decision to publish, or preparation of the manuscript.

**Competing interests:** The authors have declared that no competing interests exist.

fed mixed agriculture has proven to be resilient over generations and supported about 80% of Ethiopia's population [4].

Maize (*Zea mays* L.) is one of the world's most important food crops. Approximately 1, 162 million tons of maize are produced globally [5] with 7.7 ton/ha [6]. Production of this crop is prevalent in highland, midland, and lowland agro-ecological zones [7–9]. Nevertheless, agriculture is characterized by low-level productivity [10]. Moreover, periodic crop losses and food shortages are common phenomena [11]. Hence, food security is a challenge. Food insecurity will continue in the country due to population growth, climate change, soil fertility reduction, and young unemployment [12]. This demonstrates the importance of developing evidence-based intervention strategies.

Crop yield forecasting in Ethiopia is expensive, time-consuming, and inclined to huge errors [13]. Consequently, there are disparities between real produce and federal crop yield projections [14,15]. Late crop yield estimates, in particular, cause food aid delays, placing numerous lives in jeopardy. As a result, the country's main challenge is a lack of reliable and timely ground information. Many studies suggest that crop production estimation using remote sensing is more reliable than traditional approaches [16]. Hyperspectral remote sensing, including spectroradiometer data, provides information on crop status, health, and yield. The spectral dataset includes delicate spectral distinctive guidance on how to enhance crop variable predictive performance. It is an effective component for complete and rapid crop variable monitoring. Studies realized that there is a strong relationship between spectral vegetation indices and biomass and yield [17–19].

Accurate crop yield prediction is valuable for land users and decision makers to make strategic judgments for location-specific crop and soil management options, such as selection of crop type and fertilizer rates [20,21]. Remote sensing data are crucial in maize yield estimation [22] and crop mapping [23], and it is also necessary for irrigation, fertilization, and pest control planning [24,25]. As a result, hyperspectral data should be employed to achieve exact and geographical yield projections.

This method connects vegetation indices to in-field yield measurements at harvest time. Such a method could greatly assist the nation in allocating resources and taking prompt actions to enhance food production on time. However, the use of vegetation indices derived from spectroradiometer data to estimate crop yields at various crop growth stages has received little attention in Ethiopia. In this regard, remote sensing techniques such as visible, near-infrared, and shortwave-infrared (VIS-NIR-SWIR, 350–2500 nm wavelength) reflectance information have been employed for carefully monitoring crop growth to support appropriate agricultural development strategies [26–29]. Moreover, the spectral vegetation indices (SVIs) derived from crop reflectance were employed for estimating crop yield under various environmental conditions [30,31].

There has been minimal research into the dynamics of vegetation growth and crop productivity in Ethiopia [32], whereas the impact of variability on critical phenological dates on agricultural productivity has received scant attention [33,34]. A reliable crop monitoring system that considers not only the final yield but also the growth and development process, drivers, and application is crucial in this setting. As a result, the use of modern techniques such as spectroradiometers to obtain credible crop yield data and information for geographic and location-specific implementations should be promoted.

The findings will be useful in making management decisions such as soil management, crop selection, and fertilizer application. Furthermore, the vegetation indices-based model aids in the application of various agronomic management strategies for improved crop performance and yield, thereby closing the crop yield gap. To date, there is limited information about the assessments of the crop leaves' reflectance to evaluate the crop's GY and ABY under

rain-fed conditions. In Ethiopia, where food production remains at subsistence levels, yield estimation methodologies should be reliable to accurately represent food supply and revenue sources.

The purpose of this study is to examine spatial variability and its impact on maize yield, as well as to design a modern crop grain yield monitoring method, to improve crop yield surveys' efficiency and lower monitoring expenses. Thus, in this study, we estimate GY and ABY using leaf spectral reflectance data of maize (*Zea mays* L.) under with- and without-soil bund.

## Materials and methods

### Description of the study area

An investigation was done at Aba Gerima located in Ethiopia. The site represents midland with altitudes ranging from 1,914 to 2,121 m above sea level. Based on records from 1994 to 2021 of nearby meteorological stations, the study area receives a normal annual rainfall varying from 1,076 to 1,953 mm, with an average monthly maximum temperature of 26.99˚C and an average monthly minimum temperature of 12.58˚C (Fig 1; S1 Table). The main rainfall occurs from June to August, and the rest of the year is dry [35]. According to [36]; Acrisols, Luvisols, Vertisols, and Leptosols are the main soils in the catchment [37]. Teff (*Eragrostis tef.* (Zucc.) Trotter), finger millet (*Eleusine coracana* L.), and maize (*Zea mays* L.) are the foremost crops.

### Experimental setup and crop sampling

As per [36] guideline, we generated three slope gradients (i.e., gently sloping (2–5%), medium sloping (5.1–10%), and strongly sloping (10.1–15%)) from the cultivated lands of Aba Gerima catchment using ArcGIS software version 10.5. Twenty-four representative crop-sampling plots (CSPs) with soil bund and 24 CSPs without soil bund were identified at the study

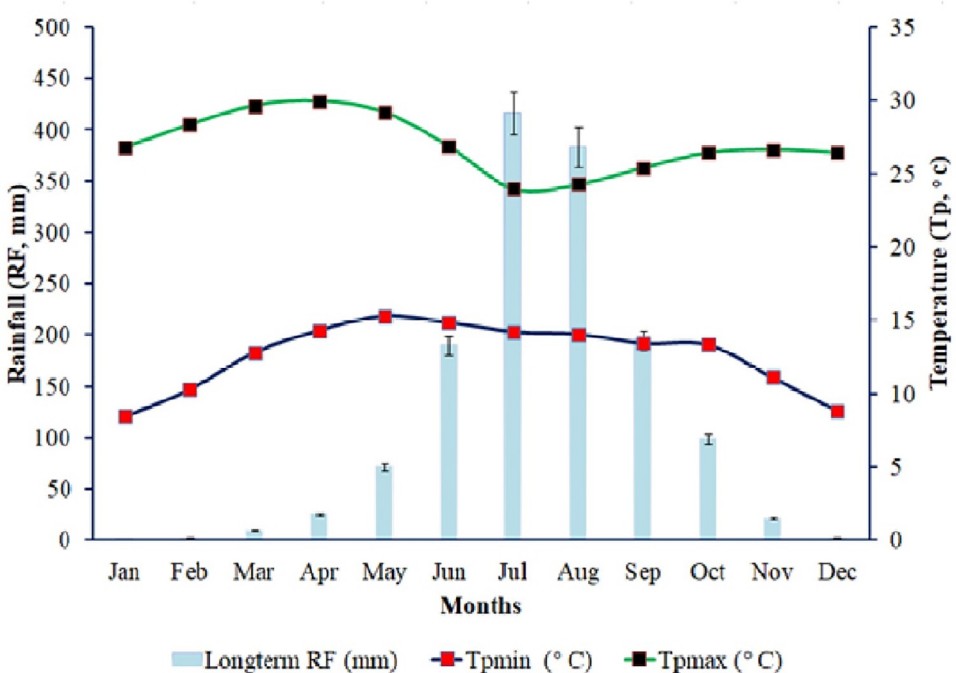

**Fig 1. Long-term (1994–2021) monthly rainfall (RF), maximum temperature (Tpmax), minimum temperature (Tpmin) in the study area.**

catchment (S2 Table). A soil bund is a soil embankment erected along a contour. As a result, a soil bund is constructed when a trench is dug and the excavated soil material is poured downward [38]. This agricultural practice one of Ethiopia's most popular physical soil and water conservation measures for reducing slope length and overland flow velocity, as well as reducing soil erosion in cropland.

The CSPs were laid out in a randomized complete block pattern. Each CSP had a minimum of 12 m x 12 m (144 m$^2$) and 15 rows replicated eight times. Following cultivation, we used row planting methods to sow hybrid breed-540 (BH540, 25 kg/ha) maize seeds. Traditional crop pest management strategies included hand weeding, early cultivation, and early planting. At sowing time, 100 kg of urea and 100 kg of di-ammonium phosphate fertilizer were used.

### Crop spectral reflectance measurement

Maize leaves from each plot were collected at the grain-filling stage of maize plants (Fig 2A) through a destructive approach (Fig 2B). The leaves were placed on a table covered with black-

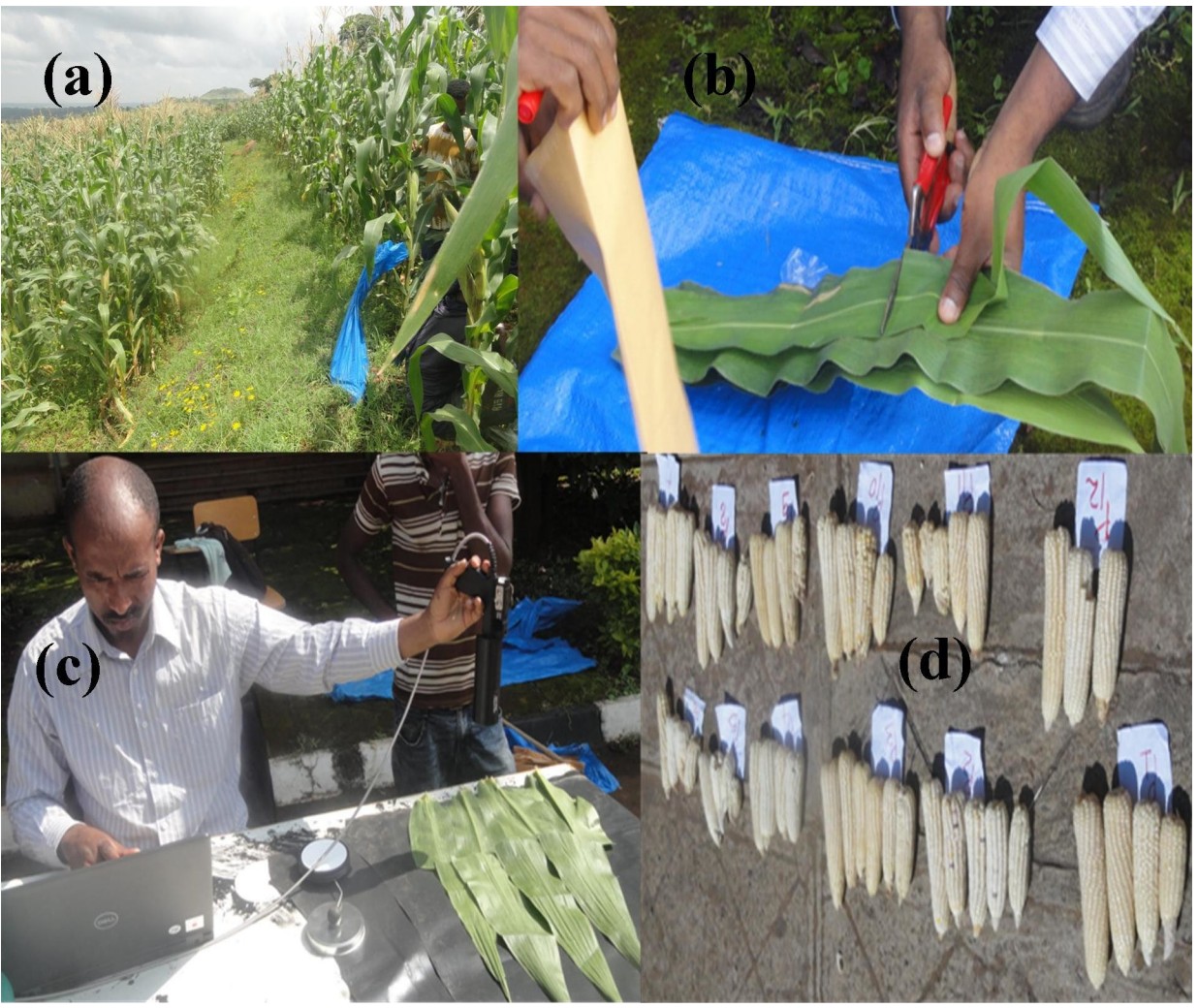

**Fig 2.** Crop sampling and reflectance measurement: (a) Maize crop sampling, (b) Sample preparation, (c) Crop reflectance measurement with a spectroradiometer, and (d) Cobs for grain weight measurement.

**Table 1. Vegetation indices (SVI) employed in the research.**

| S No | Spectral vegetation indices (SVI) | Formulae | References |
|------|-----------------------------------|----------|------------|
| 1 | Enhanced vegetation index (EVI) | $2.5 \left[ \frac{NIR-Red}{NIR+6*Red-7.5*Blue+1} \right]$ | [40] |
| 2 | Normalized difference vegetation index (NDVI) | $\frac{NIR-Red}{NIR-Red}$ | [41] |
| 3 | Green normalized difference vegetation index (GNDVI) | $\frac{R_{780}-R_{550}}{R_{780}-R_{550}}$ | [42] |
| 4 | Soil-adjusted vegetation index (SAVI) | $1.5 \left[ \frac{NIR-Red}{NIR+Red-Red+0.5} \right]$ | [43] |
| 5 | Red normalized difference vegetation index (RNDVI) | $\frac{R_{780}-R_{670}}{R_{780}-R_{670}}$ | [44] |
| 6 | Simple ratio (SR) | $\frac{R_{900}}{R_{680}}$ | [45] |

colored geo-membrane, and spectral reflectance measurement was carried out with an ASD FieldSpec IV spectroradiometer (350–2500 nm wavelength, Analytical Spectral Devices Inc., Boulder, Colorado, USA) between 10:00 and 11:00 a.m (Fig 2C).

For the determination of leaf (rather than crop canopy) level reflectance, a destructive technique was used. This enables precise control of extraneous and independent variables such as background (soil) noise, weeds, and the shade effect of the maize plant itself, among others. However, measuring leaf-level reflectance via destructive sampling is a time-consuming and costly process [39]. A white panel (Labsphere Inc., North Sutton, USA) was employed for spectroradiometer calibration. The outside light was utilized as a source of illumination. The radiometer was placed on a table covered with black colored geo-membrane.

The Remote Sensing 3 (RS³) software version 6.4 and the View Spec Pro software version 6.2 (ASD Inc, Boulder, Colorado, USA) recorded and processed the reflected spectra. The reflectance was used to compute different SVIs. Table 1 shows the description for each SVI.

## Crop yield measurement

Five random maize plants were selected, tagged in two adjacent rows, and harvested at each plot, which had homogeneous topography and plant population density, and air-dried for a week. The ABY per plot was calculated by weighing the entire plant (g/m$^2$), and converting in ton/ha. The cobs and GY of the maize samples (Fig 2D) per plot were weighed and sun-dried to calculate grain moisture content on a gravimetric base, and the yield standardized to a measured moisture content of 12%.

## Statistical analysis

Analysis of variance was utilized to evaluate the mean of GY, ABY, and SVIs with Statistical Analysis System software version 9.4 [46] and International Business Machines Corporation and Statistical Product and Service Solutions (SPSS) software version 24.0 (SPSS Inc., Chicago, IL, USA). The efficiencies of the regression models were evaluated using the coefficient of determination (R$^2$, Eq 1) and root mean squared error (RMSE, Eq 2) [47]. The R$^2$ varies from 0 to 1, with 1 being the best value [48]. As RMSE values approach zero, the model becomes a better predictor [49,50].

$$R^2 = \frac{\sum_{i=1}^{N} (\hat{y}_i - \overline{y}_i)^2}{\sum_{i=1}^{N} (y_i - \overline{y}_i)^2} \qquad\qquad Eq1$$

$$RMSE = \frac{1}{N} \sum_{i=1}^{N} (\hat{y}_i - y_i)^2 \qquad\qquad Eq2$$

**Table 2. Summary of descriptive statistics for spectral vegetation indices of maize in the Aba Gerima catchment.**

| Parameters | Min | Max | Mean (μ) ± SE | SD (σ) | CV (%) | Skewness | Kurtosis |
|---|---|---|---|---|---|---|---|
| EVI | 0.21 | 0.42 | 0.31 ± 0.01 | 0.05 | 16.13 | 0.21 | -0.82 |
| NDVI | 0.21 | 0.49 | 0.32 ± 0.01 | 0.09 | 28.13 | 0.48 | -1.24 |
| GNDVI | 0.21 | 0.39 | 0.30 ± 0.01 | 0.06 | 20.00 | 0.22 | -1.48 |
| RNDVI | 0.18 | 0.39 | 0.28 ± 0.01 | 0.05 | 17.86 | 0.30 | -0.89 |
| SR | 1.53 | 3.46 | 2.52 ± 0.08 | 0.56 | 22.22 | 0.15 | -0.94 |
| SAVI | 0.19 | 0.45 | 0.31 ± 0.01 | 0.07 | 22.58 | 0.02 | -0.83 |

Min = minimum; Max = maximum; SE, standard error of the mean; σ = standard deviation (SD); μ = mean; CV = coefficient of variation = σ/μ × 100. EVI enhanced vegetation index; NDVI normalized difference vegetation index; GNDVI green normalized difference vegetation index; RNDVI red normalized difference vegetation index; SR simple ratio; SAVI soil-adjusted vegetation index.

Where, ŷ = estimated rate; ȳ = mean observed rate; y = measured ones; N = number of observations with i = 1, 2. . . n.

## Results and discussion

### Descriptive statistics, correlation between vegetation indices, and principal component analysis

Table 2 shows a summary of the descriptive statistics of SVIs of maize. As Table 2 indicates, all spectral vegetation indices obtained a medium variation with a medium coefficient of variation (CV) ranging from 10.7 to 72.6% according to [51] guideline. The EVI, GNDVI, and RNDVI had moderate variation (CV of 12–20%), whereas NDVI, SR, and SAVI had high heterogeneity (CV of 20–30%) (Table 2). This result is in line with the findings of [52]. Data transformation was not required because the skewness and kurtosis values ranged from -2 to +2 as per reports made by [53].

We performed bivariate correlation and principal component (PC) analyses before fitting the spectral reflectance-derived vegetation indices (SVIs) into the regression models. The Pearson's correlation among maize spectral reflectance indices and the results of its interpretation are shown in Tables 3 and S3, respectively. The correlation analysis did not reveal a significantly high correlation (r > 0.8) among the SVIs at p 0.05 (Table 3). The soil-adjusted vegetation index (SAVI) had very strong positive correlations with the green normalized difference vegetation index (GNDVI). It did, however, show positive but very low correlations with the enhanced vegetation index (EVI) in the Aba Gerima catchment. However, there was no negative correlation among the SVIs in the study area (Tables 2 and S3; [54]).

The PC analysis, as shown in Table 4 and Fig 3, grouped the six SVIs into two PCs with eigenvalues greater than one that was uncorrelated with each other and cumulative variance

**Table 3. Pearson correlation matrix among spectral vegetation indices of maize.**

| | EVI | NDVI | GNDVI | RNDVI | SR | SAVI |
|---|---|---|---|---|---|---|
| EVI | 1 | 0.63 | 0.40 | 0.46 | 0.35 | 0.10 |
| NDVI | 0.63 | 1 | 0.75 | 0.50 | 0.67 | 0.50 |
| GNDVI | 0.40 | 0.75 | 1 | 0.62 | 0.79 | 0.80 |
| RNDVI | 0.46 | 0.50 | 0.618 | 1 | 0.68 | 0.42 |
| SR | 0.35 | 0.67 | 0.79 | 0.68 | 1 | 0.65 |
| SAVI | 0.10 | 0.50 | 0.81 | 0.42 | 0.65 | 1 |

EVI enhanced vegetation index; NDVI normalized difference vegetation index; GNDVI green normalized difference vegetation index; RNDVI Red normalized difference vegetation index; SR simple ratio; SAVI soil-adjusted vegetation index.

**Table 4. Variance explained by principal components and loading of the vegetation indices of maize.**

| PCs | Eigenvalues | Variance (%) | Cumulative (%) | Communalities | SVIs | PCs | |
|---|---|---|---|---|---|---|---|
| | | | | | | 1 | 2 |
| 1 | 3.84 | 63.93 | 63.93 | .91 | EVI | .036 | .951 |
| 2 | 1.03 | 17.09 | 81.02 | .78 | NDVI | .549 | .688 |
| 3 | 0.56 | 9.27 | 90.29 | .91 | GNDVI | .877 | .370 |
| 4 | 0.29 | 4.79 | 95.08 | .60 | RNDVI | .550 | .548 |
| 5 | 0.19 | 3.15 | 98.23 | .80 | SR | .811 | .382 |
| 6 | 0.106 | 1.768 | 100 | .87 | SAVI | .931 | -.025 |

PCs principal components; EVI enhanced vegetation index; NDVI normalized difference vegetation index; GNDVI green normalized difference vegetation index; RNDVI Red normalized difference vegetation index; SR simple ratio; SAVI soil-adjusted vegetation index.

greater than 70%, as also reported by [55]. The two PCs can capture the greatest number of spectral variations [56]. PC1 and PC2 explained 63.93% and 17.09% of the variance in maize yield variability in the study area, respectively. PC1 factor loadings (63.93%) and PC2 factor loadings (17.09%) best-explained maize yield variability in the catchment for 81.02% (Table 4 and Fig 3). Other research [55,56] has found that the two PCs can record the most spectrum fluctuations. As a result, EVI and SAVI contributed the most to PC1 and PC2, respectively (Fig 3). However, for further analysis, we used all spectral vegetation indices.

## Spectral reflectance response of maize leaves

The reflectivity of the maize crop between the visible and SWIR (350–2500 nm) bands is depicted in Fig 4 at the grain-filling stage. The spectral signature of the maize leaves of the different treatments is typical. However, maize plants grown in bunded plots recorded higher reflectance. The maize leaf spectral reflectance was lower in the red than NIR bands (Fig 4). This could be because of strong red absorbance by photosynthetic and plant pigments [57,58] and little NIR absorbance by cellular particles [59]. Moreover, the crop's spectral properties are differentiated, recognizable, with minimal reflectance in blue, elevated in green, very lesser in red, and quite high in the NIR [60,61]. The increased leaf greenness could be attributed to high crop intensity or total chlorophyll, linked to red absorbance and the NIR reflectance [62].

## Comparisons of yield and spectral vegetation indices

The visible (400-700nm wavelength) and near-infrared (NIR) (700-1100nm wavelength) portions are responsive and sensitive to crop genetic and morphological features [62]. Hence, the crop can take up and reflect visible and NIR radiation extra at the grain-filling stage. Furthermore, crop yield is negatively and positively associated with the spectral reflectance of red light band and near-infrared band, respectively [63]. As a result, reflectance spectroscopy methods are appropriate for offering pertinent details on crop growth parameters [64,65].

The different relationships among SVIs and maize GY recorded (Figs 5 and 6) could be associated with other factors, such as nutrient and water availability affecting crop yields. The best-fitting curve was reported between grain production and NDVI ($R^2 = 0.70$; RMSE = 0.065; P<0.0001), afterwards EVI ($R^2 = 0.65$; RMSE = 0.024; P<0.0001), while the best-fitting curve was revealed in both aboveground biomass yield and GNDVI ($R^2 = 0.77$; P<0.0001), followed by NDVI ($R^2 = 0.71$; P<0.0001). Consequently, the EVI, NDVI, and GNDVI performed best for yield estimation at the grain-filling stage. It could be due to EVI based on red, blue (450 nm), and near-infrared regions. However, the index limits and improves its sensitivity to the soil effect and high biomass areas [66].

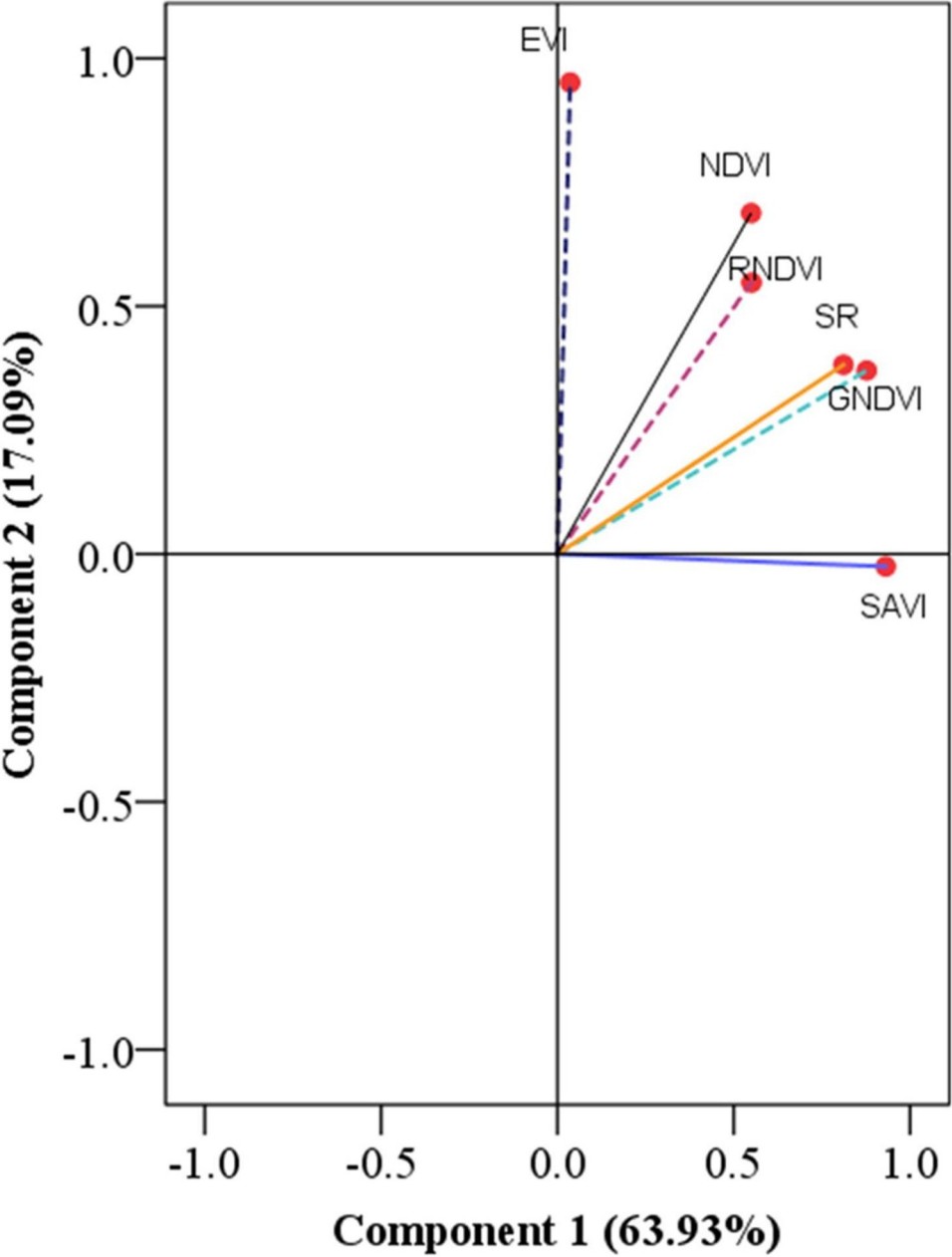

**Fig 3. Rotated components of loading of the vegetation indices of maize.** EVI enhanced vegetation index; NDVI normalized difference vegetation index; GNDVI green normalized difference vegetation index; RNDVI Red normalized difference vegetation index; SR simple ratio; SAVI soil-adjusted vegetation index.

### Spectral index of importance selection and spectral model development

Reflectance is less important than band combinations in determining crop yields [67]. As a result, different vegetation metrics like NDVI and EVI are frequently used as spatial criteria for agricultural crop productivity [68,69]. Some researchers developed an NDVI-based linear regression model to estimate maize and wheat yields [22,70,71]. The normalized difference vegetation index (NDVI) and wide dynamic range vegetation index showed the highest correlation in the maize grain yield [71,72].

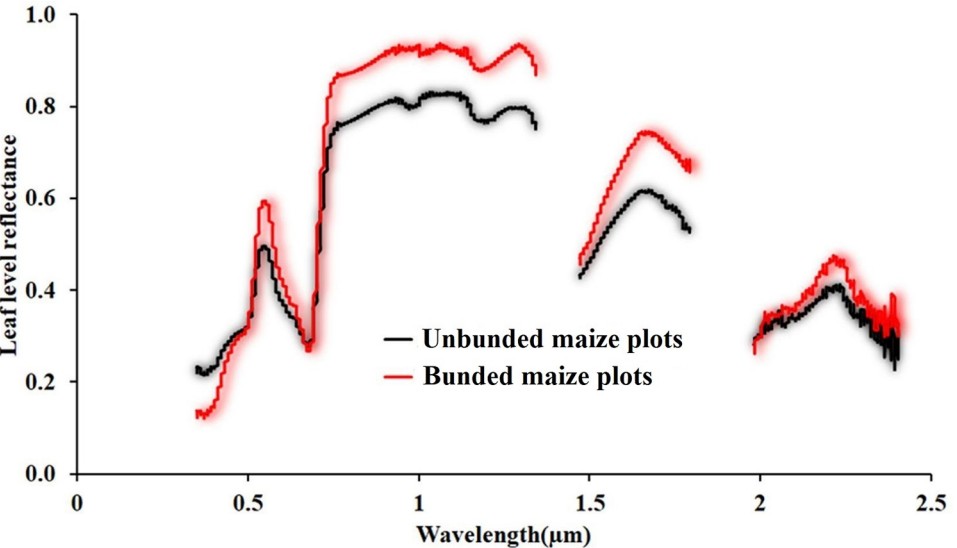

**Fig 4. Spectral reflectance response of maize leaves in Aba Gerima catchment.**

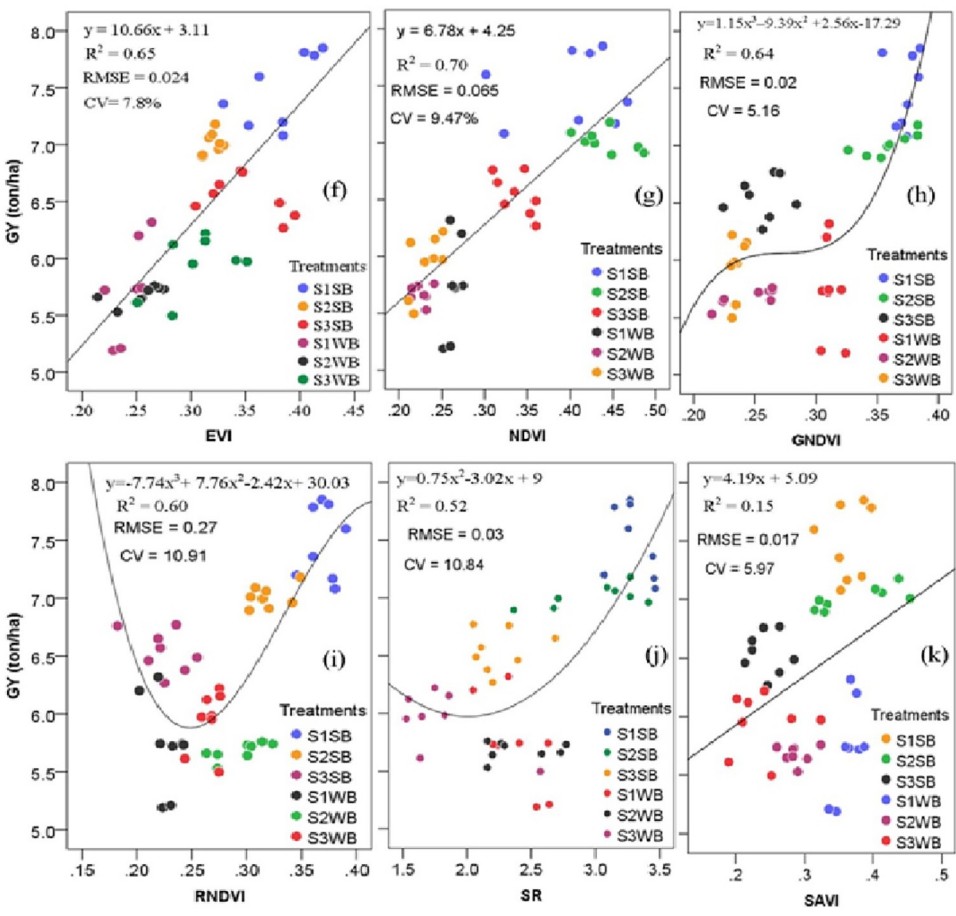

**Fig 5. Connection between spectral index values and maize grain yield measured at the crop grain-filling stage ($p < 0.01$; the total number of samples = 48).** (f) EVI, (g) NDVI, (h) GNDVI, (i) RNDVI, (j) SAVI, and (k) SR ($p < 0.01$; the total number of samples = 48). $R^2$, coefficient of determination; CV, coefficient of variation; EVI, enhanced vegetation index; GNDVI, green normalized difference vegetation index; NDVI, normalized difference vegetation index; SR, simple ratio; RNDVI, red normalized difference vegetation index; SAVI, soil adjusted vegetation index; GY, grain yield; S1, 2–5%; S2, 5–10%; S3, 10–15%; SB, soil bund; WB: Without soil bund.

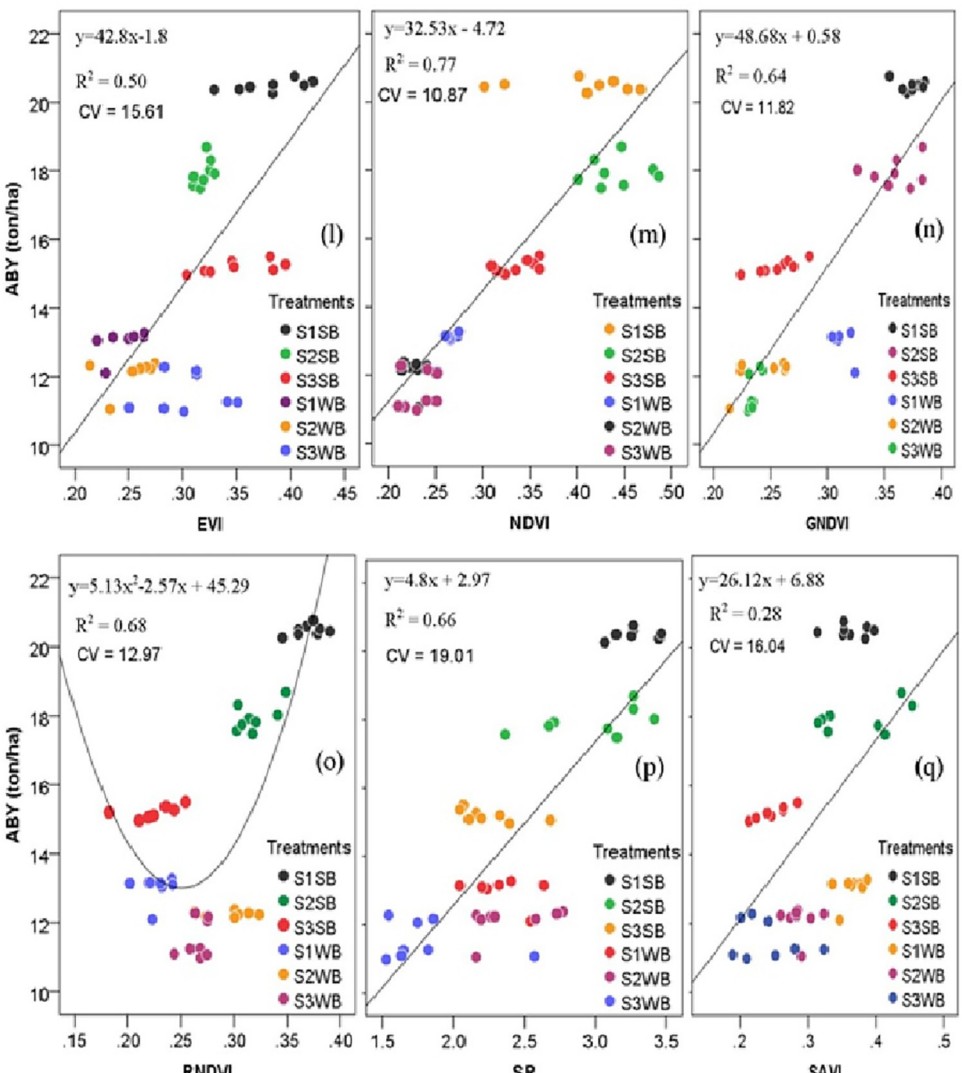

**Fig 6. Scatter plots of the measured spectral vegetation indices versus predicted GY and ABY of maize: (l) EVI, (m) NDVI, (n) GNDVI, (o) RNDVI, (p) SR, and (q) SAVI ($p < 0.01$; the number of observations = 48).** $R^2$, coefficient of determination; CV, coefficient of variation; EVI, enhanced vegetation index; GNDVI, green normalized difference vegetation index; NDVI, normalized difference vegetation index; RNDVI, red normalized difference vegetation index; SR, simple ratio; SAVI, soil adjusted vegetation index; ABY, aboveground biomass yield; S1, 2–5%; S2, 5–10%; S3, 10–15%; SB, soil bund; WB: Without soil bund.

The authors [22,73,74] found a strong association between maize grain yield and NDVI value. However, the extent of association depends on environmental conditions, varieties, crop growth stage, and agronomic practices [75]. For agricultural produce, NDVI is a superior criterion [76,77]. However, the NDVI tends to saturate as crop canopy cover increases [63]. Moreover, EVI exhibited a linear association with LAI than NDVI [37].

In addition, EVI and NDVI performed well in predicting crop grain and biomass yield with spectral response [70,78–82]. However, the low performance of RNDVI and SAVI in estimating maize yields could be related to the variation in various factors, including biotic and abiotic factors [83] and the saturation of the vegetation indices [84–86].

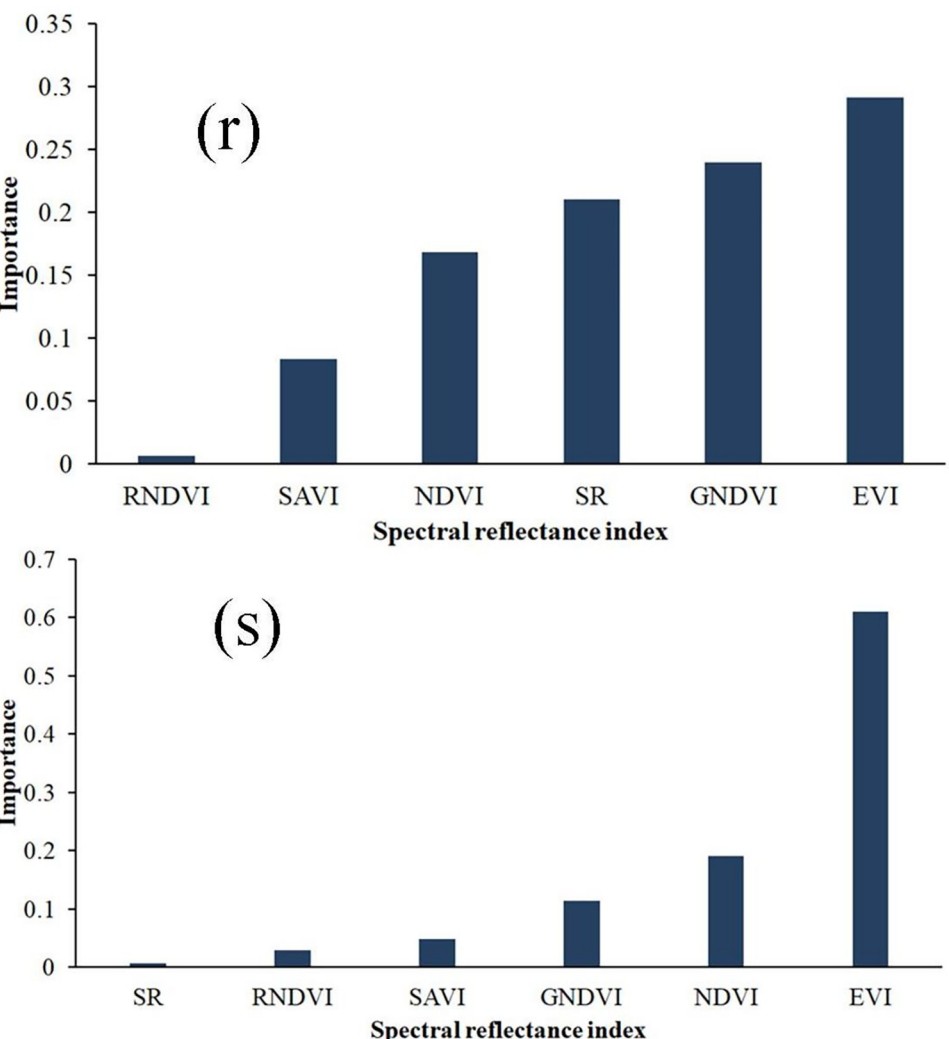

**Fig 7. The predictors' significance in the maize yield estimation.** (r) The relative importance of the predictors for the grain yield and (s) The relative importance of the predictors for the aboveground biomass; EVI, enhanced vegetation index; NDVI, normalized difference vegetation index; RNDVI, red normalized difference vegetation index; SAVI, soil adjusted vegetation index; WI, water index; GNDVI, green normalized difference vegetation index; and SR, simple ratio.

The importance of the spectral variables (EVI, NDVI, GNDVI, SAVI, RNDVI, and SR) concerning the predicted variables (GY and ABY) is shown in Fig 7. The results show that the EVI is the most important predictor for the GY (0.61) and ABY (0.29). The GNDVI-based growth metric based on crop reflectance generated at the grain-filling period had the best performance for predicting ABY based on relative importance (RI, 0.24) and coefficient of determination ($R^2$, 0.71). Using combined spectral indices, we also predicted GY with $R^2$ (0.83) and RMSE (0.24) and ABY with $R^2$ (0.78) and RMSE (0.12). The NDVI (0.19) and GNDVI (0.24) were the second-best predictors for GY and ABY. The authors [87] discovered an NDVI-based maize ABY estimation model ($R^2$ = 0.79). The GVI [88] and green chlorophyll vegetation index [89] was the most influenced maize yield variability [90]. Maize grain yield was also best predicted with GNDVI and NDVI [21,22,71,91]. Furthermore, SAVI and NDVI predicted maize yields better [86,87,92].

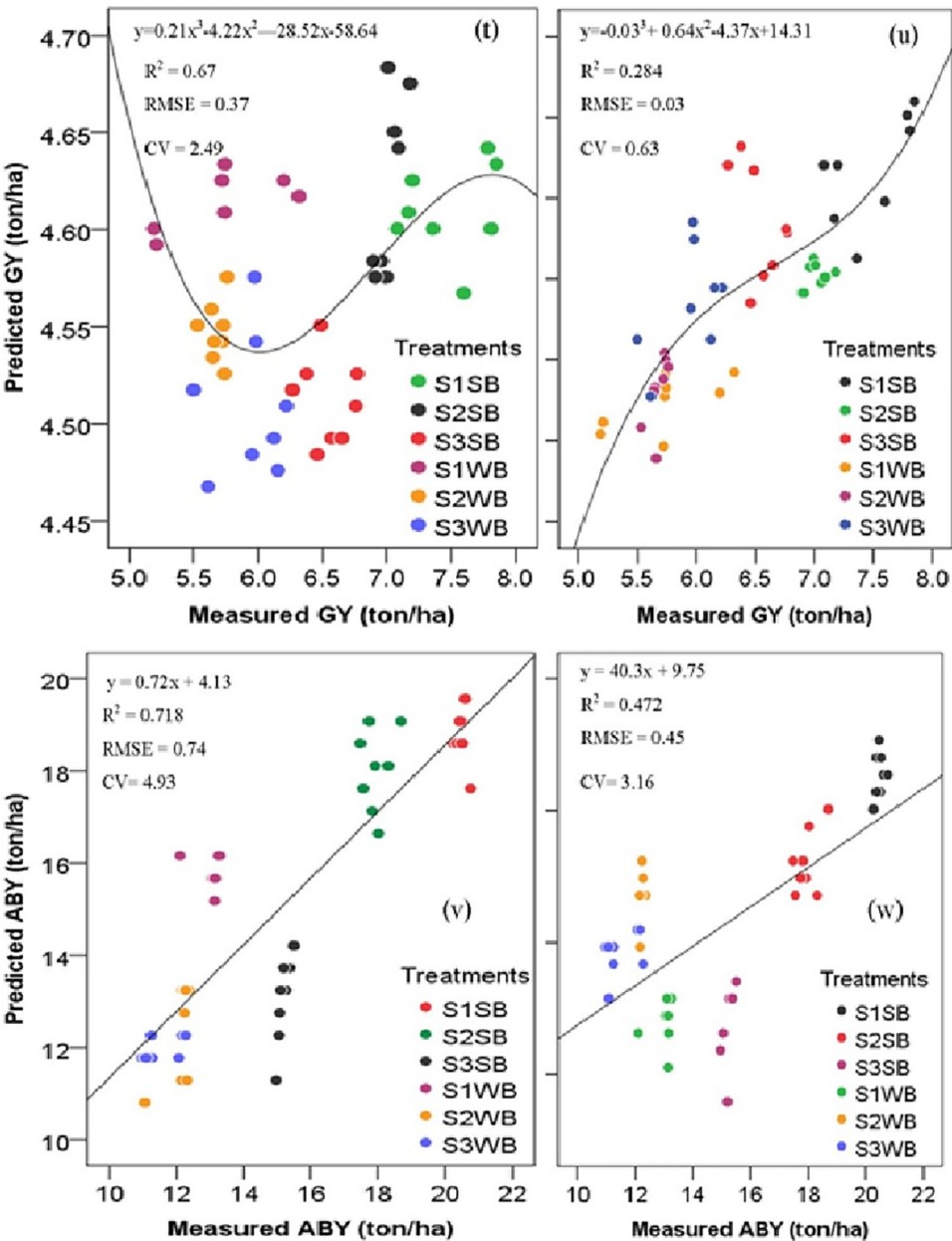

**Fig 8. Comparisons between the measured versus predicted GY and ABY of maize.** (t) GYB, (u) GYW, (v) ABYB, and (w) ABYW; R$^2$, coefficient of determination, RMSE, root mean square error; CV, coefficient of determination; GY, grain yield; ABY, aboveground biomass; S1, 2–5%; S2, 5–10%, S3, 10–15%; SB, soil bund reinforced with stone and grass, and WSB, without soil bund. (p < 0.0001; the total number of samples = 48).

The current finding with the NDVI as a parameter of significance confirmed its relevance in the chlorophyll content. This response could be due to the absorption of the red and reflected radiations in the NIR portions of the spectrum [42]. The NDVI computed from the divergence of the greatest chlorophyll absorption (red band) and the equivalent band of superior reflectance of NIR has shown achievement in maize produce prediction yield estimation

**Table 5. Model scenarios for maize grain and aboveground biomass yield in Aba Gerima catchment.**

| Treatments | GYB (ton/ha) | ABYB (ton/ha) | GYW (ton/ha) | ABYW (ton/ha) |
|---|---|---|---|---|
| S1SB | 7.18[b] | 18.71[a] | 4.61[a] | 16.54[a] |
| S1WB | 5.76[d] | 15.73[c] | 4.61[a] | 12.75[c] |
| S2SB | 6.52[ac] | 18.04[a] | 4.62[a] | 15.21[b] |
| S2WB | 5.81[d] | 12.39[b] | 4.55[b] | 14.49[d] |
| S3SB | 6.84[ab] | 13.05[b] | 4.51[b] | 12.69[c] |
| S3WB | 6.35[c] | 11.96[b] | 4.51[b] | 13.83[d] |
| Mean | 6.41 | 14.98 | 4.57 | 14.25 |
| LSD | 0.41 | 1.10 | 0.04 | 0.67 |
| CV (%) | 4.25 | 4.93 | 0.63 | 3.16 |

S1, 2–5%; S2, 5–10%; S3,10–15%; SB, soil bund reinforced with stone and grass; WSB, without soil bund; LSD, least significant difference; CV, coefficient of variation; GYB, Best grain yield; ABYB, best aboveground biomass yield; GYW, worst grain yield; ABYW, worst aboveground biomass yield. The total number of samples was 48. Letters set by different letters vary markedly (p < 0.0001), while letters preceded by the same letters would not differ substantially (p < 0.0001).

research [93]. Several studies investigated the possibility of estimating maize biomass and grain yield (GY) through VIs [21,94,95]. Furthermore, the authors [21,94] reported in a recent study that GNDVI and NDVI showed high performance in maize yield prediction from Sentinel-2 images.

The importance of the top three SVIs (EVI, NDVI, and GNDVI) could be linked to the extent of the green pigments in the maize plants. It could be due to the absorption and reflection of light at the red and NIR bands [42,96]. However, SAVI and RNDVI were the minor influential predictors in maize yield prediction [93].

The estimation approaches for maize GY, such as the SVIs employed in the current research, were effective. The highest scoring parameters were discovered. The involvement of the different SVIs to success, on the other hand, could differ depending on their interactions with the growth characters of the maize plants, including chlorophyll concentration and moisture content [71,86–87,92].

According to the values of the coefficient of determination ($R^2$), the root mean square error (RMSE), and relative importance (RI), we developed two best independent regression models that relate spectral reflectance indices (Fig 8T and 8V) to measured maize yields. Through the regression predictive models, the greatest grain yield (7.18 ton/ha) and aboveground biomass yield (18.71 ton/ha) of maize were recorded on a plot treated with soil bund on a gentle slope (Table 5; Fig 8T–8W). We obtained the maximum (7.18 ton/ha) and the minimum (4.51 ton/ha) grain yield of maize was obtained at S1SB and S3WB plots, respectively. In comparison, S1SB and S3WB provided the maximum (18.71 ton/ha) and the minimum (12.69 ton/ha) aboveground biomass yield of maize in the study area. Hence, the selected approaches could predict maize yields at its grain-filling stage based on spectral reflectance values acquired by the FieldSpec IV Spectroradiometer. However, we could achieve better predictive accuracy in crop yields if their factors are maintained across the crop cycle [90]. Thus, crop yield prediction models using spectroradiometric data are important to locate any biotic and abiotic stresses in the crop [94] and for the delineation of soil and crop management zones.

The measured GY and ABY of maize (ton/ha) were compared to their predicted values (Fig 9). The validation data were used to test the accuracy of the prediction of maize yields in the catchment using combined regression equations. The equation for the combined spectral

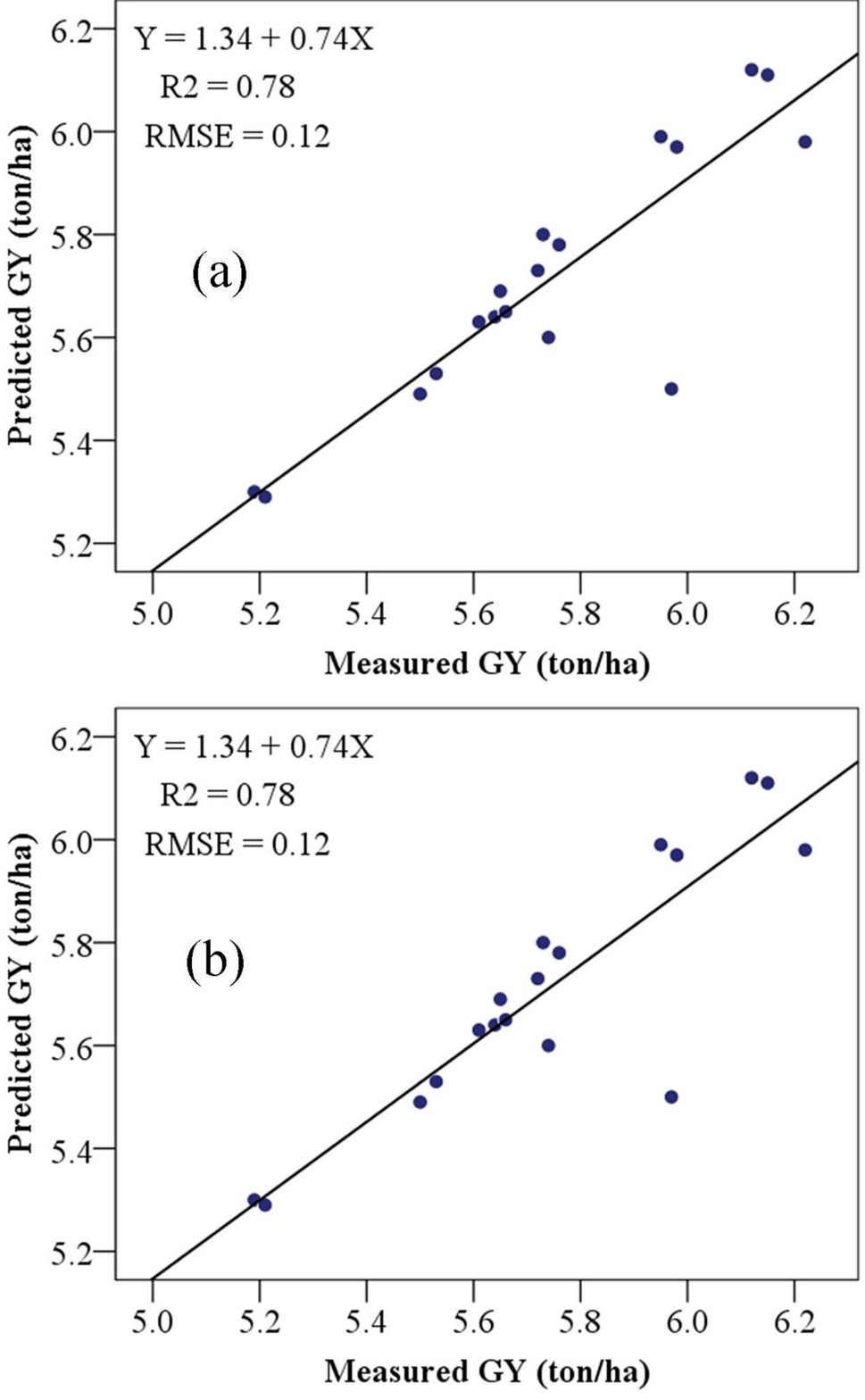

**Fig 9. Scatter graphs illustrating maize model validation results from combined models.** (a), ABY and (b), GY. ABY aboveground biomass, GY grain yield, $R^2$ coefficient of determination, and RMSE, root mean square error.

indices to predict GY with the $R^2$ (0.85) and RMSE (0.27) was:

$$GY \text{ (ton/ha)} = 2.74 + 4.8 \times EVI + 1.27 \times NDVI + 7.09 \times GNDVI - 0.72 \times SAVI + 0.33 \times RNDVI - 3.11 \times SR$$

For ABY of maize, the best fit $(R^2 = 0.97,\ RMSE = 0.56)$ was :

$$ABY \text{ (ton/ha)} = 0.61 - 14.11 \times EVI + 3.64 \times NDVI + 17.23 \times GNDVI + 15.76 \times SAVI + 1.26 \times RNDVI - 9.74 \times SR$$

Rainfall variability, soil nutrient, and moisture availability may be the key contributors to variances in expected maize output [97]. The substantial correlation between specific SVIs and maize yields could be linked to when plants reach their maximal photosynthetic capacity [98]. If maize growth factors are increased, the model's accuracy may be improved [91].

## Conclusions

A crop leaf-based spectral reflectance measurement is useful for studying crop growth and yields. The soil bund construction influenced maize's spectral vegetation indices, grain yield (GY), and aboveground biomass (ABY). The normalized difference vegetation index (NDVI)- and the enhanced vegetation index (EVI)-based growth metrics based on crop leaf reflectance had the best performance for predicting GY in terms of relative importance (RI, 0.61), coefficient of determination ($R^2$, 0.65), and small root mean square error (RMSE, 0.024 ton/ha). In-field maize yield prediction using spectroradiometer in rain-fed maize plots proved successful with spectral indices derived from leaf reflectance. However, using our data, the SR and RNDVI tend to have low efficiencies for GY and ABY.

The predictive models recorded the highest GY (7.18 ton/ha) and ABY (18.71 ton/ha) of maize at a plot treated with soil bund on a gentle slope. We obtained the maximum (7.18 ton/ha) and the minimum (4.51 ton/ha) GY of maize at S1SB and S1WB plots. In comparison, S1SB and S1WB provided the maximum (18.71 ton/ha) and the minimum (12.69 ton/ha) ABY of maize in the research region. Thus, the maize's GY and ABY can be predicted with acceptable accuracy and time using spectral reflectance indices derived from spectroradiometer in an area like the Aba Gerima catchment.

Developing and adopting a rapid and reliable crop production modeling approach could aid policy-makers in identifying yield-limiting factors, allocating resources efficiently, and implementing appropriate food initiatives to enhance food production. Furthermore, if the spectral features of the crop under several primary crop-growth stages are of future study importance, we may be able to improve the performance of the models.

## Supporting information

**S1 Table. Climatic data of Aba Gerima catchment.**
(XLSX)

**S2 Table. Geographical locations of maize sample plots at Aba Gerima catchment.**
(XLSX)

**S3 Table. Interpretation of the correlation coefficient values.**
(XLSX)

## Acknowledgments

We highly thank Anteneh Wubet, Agerselam Gualie, and Melkamu Wudu for the facilitation of our field and laboratory activities. The authors recognized the reviewers and the editors for their valuable suggestions and feedback on the early version of the paper.

## Author Contributions

**Conceptualization:** Gizachew Ayalew Tiruneh, Derege Tsegaye Meshesha, Enyew Adgo.

**Data curation:** Gizachew Ayalew Tiruneh, Derege Tsegaye Meshesha, Enyew Adgo, Atsushi Tsunekawa, Nigussie Haregeweyn, Ayele Almaw Fenta, José Miguel Reichert.

**Formal analysis:** Gizachew Ayalew Tiruneh, Enyew Adgo, José Miguel Reichert.

**Funding acquisition:** Gizachew Ayalew Tiruneh, Derege Tsegaye Meshesha, Atsushi Tsunekawa, Ayele Almaw Fenta.

**Investigation:** Nigussie Haregeweyn, Ayele Almaw Fenta, José Miguel Reichert.

**Methodology:** Gizachew Ayalew Tiruneh, Derege Tsegaye Meshesha, Atsushi Tsunekawa, Nigussie Haregeweyn, Ayele Almaw Fenta, José Miguel Reichert.

**Project administration:** Atsushi Tsunekawa, Nigussie Haregeweyn, Ayele Almaw Fenta.

**Resources:** Ayele Almaw Fenta.

**Supervision:** Atsushi Tsunekawa.

**Writing – original draft:** Gizachew Ayalew Tiruneh.

**Writing – review & editing:** Gizachew Ayalew Tiruneh, Derege Tsegaye Meshesha, Enyew Adgo, Atsushi Tsunekawa, Nigussie Haregeweyn, Ayele Almaw Fenta.

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
