## [Decision Letter · Decision Letter 0]

7 Mar 2022

PONE-D-22-01589A leaf reflectance-based crop yield modeling in Northwest EthiopiaPLOS ONE

Dear Dr. Tiruneh,

Thank you for submitting your manuscript to PLOS ONE. After careful consideration, we feel that it has merit but does not fully meet PLOS ONE’s publication criteria as it currently stands. Therefore, we invite you to submit a revised version of the manuscript that addresses the points raised during the review process.

A more contextualized introduction is required that provides a strong background for the study. This may include such information as general overview of the state of agriculture and food security in Ethiopia, a brief explanation of why crop yield estimation is important in agriculture/food security and stronger justification for remote sensing based assessments. There is need for a more detailed description of the study design including the experimental settings for the study. There is also a general need to improve the scientific writing by thoroughly reviewing the manuscript or sending it for english language editing.

We look forward to receiving your revised manuscript.

Kind regards,

Abel Chemura

Academic Editor

PLOS ONE

Journal Requirements:

"The research was funded by the Science and Technology Research Partnership for Sustainable Development (grant number JPMJSA1601), Japan Science and Technology Agency/Japan International Cooperation Agency (JICA). We highly thank Anteneh Wubet, Agerselam Gualie, and Melkamu Wudu for the facilitation of our field and laboratory activities. The authors recognized the reviewers and the editors for their valuable suggestions and feedback on the early version of the paper."

We note that you have provided funding information. However, funding information should not appear in the Acknowledgments section or other areas of your manuscript. We will only publish funding information present in the Funding Statement section of the online submission form. 

"The research was funded by the Science and Technology Research Partnership for Sustainable Development (grant number JPMJSA1601), Japan Science and Technology Agency/Japan International Cooperation Agency (JICA). Gizachew Ayalew received the fund award. The funders had no role in study design, data collection and analysis, decision to publish, or preparation of the manuscript."

6. We note that you have referenced (4. Greatrex H. The Application of Seasonal Rainfall Forecasts and Satellite Rainfall Estimates to Seasonal Crop Yield Forecasting for Africa. Unpublished PhD Thesis, University of Reading, UK. 2012.) which has currently not yet been accepted for publication. Please remove this from your References and amend this to state in the body of your manuscript: (4. Greatrex H. The Application of Seasonal Rainfall Forecasts and Satellite Rainfall Estimates to Seasonal Crop Yield Forecasting for Africa. Unpublished PhD Thesis, University of Reading, UK. 2012. [Unpublished]”) as detailed online in our guide for authors

7. We note that Figure 1 in your submission contain map image which may be copyrighted. All PLOS content is published under the Creative Commons Attribution License (CC BY 4.0), which means that the manuscript, images, and Supporting Information files will be freely available online, and any third party is permitted to access, download, copy, distribute, and use these materials in any way, even commercially, with proper attribution. For these reasons, we cannot publish previously copyrighted maps or satellite images created using proprietary data, such as Google software (Google Maps, Street View, and Earth). For more information, see our copyright guidelines: http://journals.plos.org/plosone/s/licenses-and-copyright.

Reviewers' comments:

Reviewer's Responses to Questions

**Comments to the Author**

1. Is the manuscript technically sound, and do the data support the conclusions?

Reviewer #1: Yes

Reviewer #2: No

2. Has the statistical analysis been performed appropriately and rigorously? 

Reviewer #1: No

Reviewer #2: Yes

3. Have the authors made all data underlying the findings in their manuscript fully available?

Reviewer #1: No

Reviewer #2: Yes

4. Is the manuscript presented in an intelligible fashion and written in standard English?

Reviewer #1: Yes

Reviewer #2: No

5. Review Comments to the Author

Reviewer #1: The manuscript is technically sound. However, there is room for further improvement. The statistical analysis can be improved as well. I suggest sending the paper for English editing to improve the overall strength of the manuscript. All the comments have been included in the attachment.

Reviewer #2: I found some serious deficiencies in the way the article is written. The authors did not specify the experimental design used. Data collection protocols are not described in sufficient details. For example, Fig. 4 shows the spectral data being taken on leaves laid out on a ‘bench/table’ but in the text there is no mention of destructive (Fig 4b and c) sampling. Why the authors chose destructive sampling is not justified in the article. I would expect the researchers to collect spectral data in-situ in the field. However, whatever the method used should have been spelt out clearly and provide justification. How did they control the background noise? I do not see any value whatsoever in showing readers a heap of harvested maize stover (Fig. 4d) and ear/cob images (Fig.4e, unless there was cob/ear imaging as part of data collection, which I did not see in the article). Averaging spectral signatures of different treatments is not advisable (Fig. 5) unless the authors found no significant differences across treatments (which should be stated). Otherwise, I see no value in Fig 5. The authors included supplementary data (S1 & S2) but this is not mentioned anyway in the article except Table notations (L403-405). I recommend major revision. See specific comments on attached document.

6. PLOS authors have the option to publish the peer review history of their article (what does this mean?). If published, this will include your full peer review and any attached files.

Reviewer #1: No

Reviewer #2: No

---

## [Author Response · Author response to Decision Letter 0]

25 Mar 2022

Date: March 14, 2022

Rebuttal letter

PONE-D-22-01589

We are happy about the academic editor and the reviewers’ comments, which strengthen the current version of the manuscript “A leaf reflectance-based crop yield modeling in Northwest Ethiopia”. In addition, our supreme sincere gratitude goes to you and the reviewers who devote their valuable time to bring our manuscript to a competent paper. 

We have provided a one by one reply to all concerns and comments given below. We thank you for your consideration of this resubmission and look forward to your response.

Best regards,

Gizachew Ayalew Tiruneh (on behalf of all co-authors)

Lecturer in Debre Tabor University

Ph.D. Fellow in soil science, Bahir Dar University

Email: tiruneh1972@gmail.com

 

Dear editor and reviewers, thank you so much for taking your valuable time to elevate the quality of our manuscript. We do hope that the editor’s and Reviewer’s concerns will be addressed.

Editor comments:

Comment 1: A rebuttal letter that responds to each point raised by the academic editor and reviewer(s). You should upload this letter as a separate file labeled 'Response to Reviewers'.

Response: We addressed the concerns provided by the editor and reviewers and uploaded a file labeled “Response to Reviewers”.

Comment 2: A marked-up copy of your manuscript that highlights changes made to the original version. You should upload this as a separate file labeled 'Revised Manuscript with Track Changes'.

Response: We tried to highlight our revised paper with tracked changes. We uploaded this as a separate file labeled 'Tracked changes'.

Comment 3: An unmarked version of your revised paper without tracked changes. You should upload this as a separate file labeled 'Manuscript'.

Response: We revised our manuscript without tracked changes. We uploaded this as a separate file labeled 'Manuscript'.

Comments 4: Response: We have not made any changes to financial disclosure.

Reviewer comments:

Reviewer 1:

Comment 1: The introduction section should be re-written, it lacks depth. Currently, it sounds like highlights of a paper. The authors should motivate on the relevance of their study in improving food security. The objective (s) of the study are not adequately conceptualized in the introduction. The following should be included in this section

• General overview of the state of agriculture and food security in Ethiopia (not a few random statistics)

• Give a brief explanation of why crop yield estimation is important in agriculture/food security? Why should we do it? If there is no adequate justification on this issue, the study won’t be necessary 

• Give an account of the old/traditional methods that were being used to estimate crop yields…what are their shortcomings/weaknesses ---- at this point remote sensing comes in

• Introduce remote sensing, how does it address the weakness presented by traditional methods in crop yield estimation - Mention previous works (hyperspectral remote sensing preferably, since you are using a Spectroradiometer). 

• Introduce spectral vegetation indices, what are their significance in building crop yield estimation models? 

• Now that we know remote sensing has been previously used in crop yield estimation, please highlight the gap that you would like to address – what new information do you wish to generate?

Response: Thank you. We appreciate your valuable comments. We tried to address the comments and incorporated them in the revised manuscript in the introduction section. We hope that this revised version will be satisfying.

Comment 2. The slope gradients on the experimental design are overlapping. If gently sloping is 2-5% and 5-10% is slopping then how would we classify a slope with 5%, which category does it fit in? 

Response: Thank you for your concern. As per FAO (2006) guideline, we generated three slope gradients (i.e., gently sloping (2–5 %), sloping (5–10 %), and strongly sloping (10–15 %)). This classification has a problem of rounding a number to the nearest digit. As result, gently sloping (2–5 %) means slopes greater that 2% and less than or equal to 5%. Class 5-10% means slopes greater that 5% and less than or equal to 10%. And strongly sloping (10–15 %) means slopes greater that 5% and less than or equal to 10%.

Comment 3: Figure 3 is not adding any value to the manuscript. The focus is not on the ASD but crop yield modeling. I suggest you remove figure 3. 

Response: Thank you for your suggestion. I deleted figure 3 (old manuscript).

Comment 4. There is a lot of relevant information missing in the methodology section; five maize plants were harvested at each plot and air-dried for a week’. – How many plots did you have? 

Response: Twenty-four representative crop-sampling plots (CSPs) with soil bund and 24 CSPs without soil bund were used for this study. 

Comment 5. water content of 12 % - how did we get to the 12% water content. Give a scientific explanation/justification, add references as well. 

Response: The moisture content of maize seed was measured with a hand-held grain moisture tester (model AG-12, A-Grain, India). Grain per plot (kg) adjusted for grain moisture content at harvest.

Comment 6. There are many spectral vegetation indices used in remote sensing studies. However, the current study chose six indices. How did you choose them? What is the justification for using those six indices? 

Response: Thank you. The two regions of the spectrum (visible red and near infra-red) are photosyntheticaly active spectrum bands for most plants. Hence, they are incorporated in defining most vegetation indices under the current study. Moreover, these indices perform very well for assessing performance of crops in terms of growth, biomass, and grain yield of most crops grown within a field (Jackson and Huete, 1991; Wiegand et al., 1991; Bastiaanssen, 2003). We hope that this revised version will be satisfying.

Comment 7. ‘Fig. 5 shows the spectral response of the maize crop between the visible and SWIR (350- 128 2500 nm) bands over the growing season’. The authors mention that the spectral response curve was over the growing season. However, the explanation in the results section does not specify the phenological stage at which the spectral reflectance was measured. If the spectral measurements were recorded over the growing season as expected, then we expect to see spectral response curves at each growth stage (not sure how many for maize). 

Response: We share your concern. The spectral response of the maize crop between the visible and SWIR (350- 128 2500 nm) bands was examined at the grain-filling stage’ However, the crop responded differently to radiation with- and without-bund (Fig. 4, the revised manuscript). We hope that this revised version will be satisfying.

Reviewer 2:

Comment 1: I found some deficiencies in the way the article is written. The authors did not specify the experimental design used. Data collection protocols are not described in sufficient details. For example, Fig. 4 shows the spectral data being taken on leaves laid out on a ‘bench/table’ but in the text there is no mention of destructive (Fig 4b and c) sampling. Why the authors chose destructive sampling is not justified in the article. I would expect the researchers to collect spectral data in-situ in the field. However, whatever the method used should have been spelt out clearly and provide justification. How did they control the background noise? 

Response: Thank you. An experiment was established and the CSPs were laid out in a randomized complete block design. Each CSP has a minimum of 12 m × 12 m (144 m2) replicated eight times and 15number of rows. After cultivation, we sowed the maize seeds of hybrid breed-540 (25 kg/ha) by row planting methods. Crop pests were controlled with traditional methods, including early cultivation, early planting, planting other plants at border of the crop fields, and so on. About 100 kg/ha urea and 100 kg/ha di-ammonium phosphate fertilizer were also applied at sowing time. 

For the determination of leaf (rather than crop canopy) level reflectance, a destructive technique was used. This enables precise control of extraneous and independent variables such as background (soil) noise, weeds, the shade effect of the maize plant itself, and so on. However, measuring leaf level reflectance via destructive sampling is a time-consuming and costly process (Coops et al., 2003). A white panel (Labsphere Inc., North Sutton, USA) was employed for spectroradiometer calibration. Sun radiation at open air (outside) was used as a light source. The radiometer was placed on a table covered with black coloured geo-membrane. 

Five random maize plants was selected, tagged in two adjacent rows, and harvested at each plot, which has homogeneous topography and plant population density and air-dried for a week. ABY per plot was calculated by balancing the entire plant, gauging it with a balance, and conveying in ton/ha. The cobs and GY of the maize samples per plot were weighed and sun-dried to calculate grain moisture content on a gravimetric model and the yield standardized to a measured moisture content of 12 % was articulated as a ton/ha. The above comments tried to address and incorporate them in the revised manuscript in the methodology section. We hope that this revised version will be satisfying.

Comment 2. I do not see any value whatsoever in showing readers a heap of harvested maize stover (Fig. 4d) and ear/cob images (Fig.4e, unless there was cob/ear imaging as part of data collection, which I did not see in the article). 

Response: Thank you for your suggestion. We removed Fig. 4d (old manuscript) and maintained ear/cob images (Fig.4e; old manuscript) as they were used to measure grain weight and moisture content.

Comment 3. Averaging spectral signatures of different treatments is not advisable (Fig. 5) unless the authors found no significant differences across treatments (which should be stated). Otherwise, I see no value in Fig 5. 

Response: Thank you for your suggestion. We share your concern. We included the spectral responses of maize plants grown in bunded- and non-bunded plots. This helps to show the influence of constructing bunds on reflectance, vegetation indices, and crop yields. We hope that this revised version will be satisfying.

Comment 4. The authors included supplementary data (S1 & S2) but this is not mentioned anyway in the article except Table notations (L403-405). 

Response: We cited the ssupplementary tables S1 table and S2 Tables in revised manuscript’s text.

Comment 5. Introduction

• Revise L41

• The sentence (L52-53) ‘…examined …’ need to be revised.

Response: Thank you. We have revised the introduction part of the study.

Comment 6. The methodology is not described in sufficient detail.

• L94-95. I do not see the value of Figure 4d ‘maize heap’ in this article

• L103 Five plants were harvested. How? From which row? When (days after planting/phenological stage)?

Response: Thank you. We also have re-structured the Methodology section. 

Comment 7. Results and discussion

• Figure 5 shows the spectral signature of maize. At what phenological stage of the maze was this measured. From M&M, 24 plots were planted with or without soil bund and at 3 different slopes. Does Fig. 5 represent the average of all the different treatments? What value does Fig. 5 add to the manuscript if it is an average across treatments? I would expect the authors to state concisely what was measured, when and a comparison of the different treatment used in the experiment. 

• Results displayed in Figs 6-7 (old manuscript) need to be well explained and discussed. This is lacking in L138-151

Response: Thank you for the comments. We tried to incorporate the comments in the results and discussion section of the study. Note: Specific comments raised by both reviewers were also addressed and incorporated in the revised manuscript.

Additional comments

Comment 1: When submitting your revision, we need you to address these additional requirements. Please ensure that your manuscript meets PLOS ONE's style requirements, including those for file naming. The PLOS ONE style templates can be found at 

Response: Thank you. We tried to follow the PLOS ONE's style requirements throughout the manuscript.

Comment 2. We suggest you thoroughly copyedit your manuscript for language usage, spelling, and grammar. If you do not know anyone who can help you do this, you may wish to consider employing a professional scientific editing service. The name of the colleague or the details of the professional service that edited your manuscript. A copy of your manuscript showing your changes by either highlighting them or using track changes (uploaded as a *supporting information* file). A clean copy of the edited manuscript (uploaded as the new *manuscript* file)

Response: Thank you. We have thoroughly revised our manuscript with the help of Grammarly (premium) and Turnitin software, and we do hope that the concerns will be addressed. 

Comment 3. Please provide additional details regarding participant consent. In the ethics statement in the Methods and online submission information, please ensure that you have specified (1) whether consent was informed and (2) what type you obtained (for instance, written or verbal, and if verbal, how it was documented and witnessed). If your study included minors, state whether you obtained consent from parents or guardians. If the need for consent was waived by the ethics committee, please include this information. If you are reporting a retrospective study of medical records or archived samples, please ensure that you have discussed whether all data were fully anonymized before you accessed them and/or whether the IRB or ethics committee waived the requirement for informed consent. If patients provided informed written consent to have data from their medical records used in research, please include this information

Response: Unfortunately, it is not applicable for this study. Hence, we removed the ethical statement from the revised manuscript.

Comment 4. Thank you for stating the following in the Acknowledgments Section of your manuscript: "The research was funded by the Science and Technology Research Partnership for Sustainable Development (grant number JPMJSA1601), Japan Science and Technology Agency/Japan International Cooperation Agency (JICA). We highly thank Anteneh Wubet, Agerselam Gualie, and Melkamu Wudu for the facilitation of our field and laboratory activities. The authors recognized the reviewers and the editors for their valuable suggestions and feedback on the early version of the paper." We note that you have provided funding information. However, funding information should not appear in the Acknowledgments section or other areas of your manuscript. We will only publish funding information present in the Funding Statement section of the online submission form. 

"The research was funded by the Science and Technology Research Partnership for Sustainable Development (grant number JPMJSA1601), Japan Science and Technology Agency/Japan International Cooperation Agency (JICA). Gizachew Ayalew received the fund award. The funders had no role in study design, data collection and analysis, decision to publish, or preparation of the manuscript." Please include your amended statements within your cover letter; we will change the online submission form on your behalf.

Response: Thank you. As per the comments, we improved the cover letter.

Comment 5. We note that you have stated that you will provide repository information for your data at acceptance. Should your manuscript be accepted for publication, we will hold it until you provide the relevant accession numbers or DOIs necessary to access your data. If you wish to make changes to your Data Availability statement, please describe these changes in your cover letter and we will update your Data Availability statement to reflect the information you provide.

Response: Data required for this study are within the manuscript and/or supplementary files. If other data are needed, we are happy to provide it upon request. Hence, we updated the cover letter as per the comment.

Comment 6. We note that you have referenced (4. Greatrex H. The Application of Seasonal Rainfall Forecasts and Satellite Rainfall Estimates to Seasonal Crop Yield Forecasting for Africa. Unpublished PhD Thesis, University of Reading, UK. 2012.) which has currently not yet been accepted for publication. Please remove this from your References and amend this to state in the body of your manuscript: (4. Greatrex H. The Application of Seasonal Rainfall Forecasts and Satellite Rainfall Estimates to Seasonal Crop Yield Forecasting for Africa. Unpublished PhD Thesis, University of Reading, UK. 2012. [Unpublished]”) as detailed online in our guide for authors

Response: Thank you for your suggestion and we replaced it with the published version in the revised manuscript as: Greatrex HL, Grimes DI, Wheeler TR. Application of seasonal rainfall forecasts and satellite rainfall observations to crop yield forecasting for Africa. In EGU General Assembly Conference Abstracts 2009 Apr (p. 5434).

Comment 7. We note that Figure 1 in your submission contain map image which may be copyrighted. All PLOS content is published under the Creative Commons Attribution License (CC BY 4.0), which means that the manuscript, images, and Supporting Information files will be freely available online, and any third party is permitted to access, download, copy, distribute, and use these materials in any way, even commercially, with proper attribution. For these reasons, we cannot publish previously copyrighted maps or satellite images created using proprietary data, such as Google software (Google Maps, Street View, and Earth). For more information, see our copyright guidelines: http://journals.plos.org/plosone/s/licenses-and-copyright.

“I request permission for the open-access journal PLOS ONE to publish XXX under the Creative Commons Attribution License (CCAL) CC BY 4.0 

(http://creativecommons.org/licenses/by/4.0/). Please be aware that this license allows unrestricted use and distribution, even commercially, by third parties. Please reply and provide explicit written permission to publish XXX under a CC BY license and complete the attached form.”

The Gateway to Astronaut Photography of Earth (public 

domain): http://eol.jsc.nasa.gov/sseop/clickmap/

Response: Thank you for your suggestion and we removed the Fig. 1d from the revised manuscript. However, we downloaded and computed the land-use/land- cover classes of the study catchment from Sentinel-2 satellite image, which is freely available from USGS website. 

 

Reviewer's Responses to Questions

Comments to the Author

1. Is the manuscript technically sound, and do the data support the conclusions?

Reviewer #1: Yes

Reviewer #2: No

Response: Thank you. We have gone thoroughly the revised manuscript, and hopefully that the second Reviewer will be satisfied.

2. Has the statistical analysis been performed appropriately and rigorously?

Reviewer #1: No

Reviewer #2: Yes

Response: Thank you. We have gone thoroughly the revised manuscript, and hopefully that the first Reviewer will be satisfied.

3. Have the authors made all data underlying the findings in their manuscript fully available?

Reviewer #1: No

Reviewer #2: Yes

Response: Thank you. Data required for this study are within the manuscript and/or supplementary files. If other data are needed, we are happy to provide it upon request.

4. Is the manuscript presented in an intelligible fashion and written in standard English?

Reviewer #1: Yes

Reviewer #2: No

Response: Thank you for your advice. We have thoroughly revised our manuscript with the help of Grammarly (premium) and Turnitin software, and we do hope that the second reviewer’s concerns will be addressed.

5. Review Comments to the Author

Reviewer #1: The manuscript is technically sound. However, there is room for further improvement. The statistical analysis can be improved as well. I suggest sending the paper for English editing to improve the overall strength of the manuscript. All the comments have been included in the attachment.

Response: Thank you. We have thoroughly revised our manuscript with the help of Grammarly (premium) and Turnitin software, and we do hope that the first reviewer’s concerns will be addressed.

Reviewer #2: I found some serious deficiencies in the way the article is written. The authors did not specify the experimental design used. Data collection protocols are not described in sufficient details. For example, Fig. 4 shows the spectral data being taken on leaves laid out on a ‘bench/table’ but in the text there is no mention of destructive (Fig 4b and c) sampling. Why the authors chose destructive sampling is not justified in the article. I would expect the researchers to collect spectral data in-situ in the field. However, whatever the method used should have been spelt out clearly and provide justification. How did they control the background noise? I do not see any value whatsoever in showing readers a heap of harvested maize stover (Fig. 4d) and ear/cob images (Fig.4e, unless there was cob/ear imaging as part of data collection, which I did not see in the article). Averaging spectral signatures of different treatments is not advisable (Fig. 5) unless the authors found no significant differences across treatments (which should be stated). Otherwise, I see no value in Fig 5. The authors included supplementary data (S1 & S2) but this is not mentioned anyway in the article except Table notations (L403-405). I recommend major revision. See specific comments on attached document.

Response: Thank you. We have gone thoroughly the revised manuscript, and hopefully that the second reviewer will be satisfied.

6. PLOS authors have the option to publish the peer review history of their article (what does this mean?). If published, this will include your full peer review and any attached files.

Do you want your identity to be public for this peer review? For information about this choice, including consent withdrawal, please see our Privacy Policy.

Reviewer #1: No

Reviewer #2: No

Response: Thank you. We have used PACE with this submission, so this should be right.

Please note that once again, thank you very much. Your comments are greatly appreciated.

Best regards,

Gizachew Ayalew Tiruneh (on behalf of all co-authors)

Lecturer in Debre Tabor University

Ph.D. Fellow in soil science, Bahir Dar University, Email: tiruneh1972@gmail.com

---

## [Decision Letter · Decision Letter 1]

16 May 2022

PONE-D-22-01589R1A leaf reflectance-based crop yield modeling in Northwest EthiopiaPLOS ONE

Dear Dr. Tiruneh,

Thank you for submitting your manuscript to PLOS ONE. After careful consideration, we feel that it has merit but does not fully meet PLOS ONE’s publication criteria as it currently stands. Therefore, we invite you to submit a revised version of the manuscript that addresses the points raised during the review process.

While notable improvements have been made from the original submission, there are some revisions that remain pending that the authors should attend to in the methods and the discussion, particularly to ensure that all key results are discussed.

We look forward to receiving your revised manuscript.

Kind regards,

Abel Chemura

Academic Editor

PLOS ONE

Journal Requirements:

Reviewers' comments:

Reviewer's Responses to Questions

**Comments to the Author**

1. If the authors have adequately addressed your comments raised in a previous round of review and you feel that this manuscript is now acceptable for publication, you may indicate that here to bypass the “Comments to the Author” section, enter your conflict of interest statement in the “Confidential to Editor” section, and submit your "Accept" recommendation.

Reviewer #1: All comments have been addressed

Reviewer #2: (No Response)

2. Is the manuscript technically sound, and do the data support the conclusions?

Reviewer #1: Yes

Reviewer #2: Yes

3. Has the statistical analysis been performed appropriately and rigorously? 

Reviewer #1: Yes

Reviewer #2: No

4. Have the authors made all data underlying the findings in their manuscript fully available?

Reviewer #1: Yes

Reviewer #2: Yes

5. Is the manuscript presented in an intelligible fashion and written in standard English?

Reviewer #1: Yes

Reviewer #2: No

6. Review Comments to the Author

Reviewer #1: Introduction

Page 3 line 45: Replace 'time taking' with 'time consuming'

Page 3 line 48 and 49: The two sentences on these lines begin with 'As a result...' consider revising one of the sentences.

Generally, the motivation in the introduction is still weak. The motivation should be strengthened.

Reviewer #2: Reviewer Comments

Summary

The authors made partial corrections suggested in the first round of review. The manuscript still lacks details, mainly in the following areas: motivation/rationale in the introduction; English editing; clarity in materials and methods; further analysis using a combination of VIs in the model; referencing latest publications on yield modeling using remote sensing; and detailed discussion of results. Specific comments are given below. I recommend further revision.

Abstract

The last concluding statement (L35-37) talks about policy makers effectively managing resources! How?

Key words:

Why is “Arid” included as a key word when it is not mentioned anyway in the manuscript body except authors’ address and references? The study area receives rainfall ranging from 1,076 to 1,953 mm and cannot be classified as arid.

Introduction

The introduction needs to be revised. The motivation/rationale is still weak or not convincing.

L49-51 should be taken to the second last paragraph of introduction and remove “As a result” from the statement.

L61-62 need rephrasing and qualifying.

L64 remove “as essential devices”

L75-76 remove “examined to”

L77, authors need to define the term “bund” for readers who are not familiar with the term/system

Material and Methods

The authors mentioned “midland”. This is not the right way to classify based on altitude. Better to say mid-altitude. “Midland” may mean a different thing altogether.

L86, the word “Teff” should not be in italics

The slope category is overlapping (L92-93): gentle (2–5 %), sloping (5–10 %), and strongly sloping (10–15 %). Please correct. I also suggest the mid slope (5-10%) category to be referred to as “medium” sloping rather than just “sloping” with corrections on overlapping categories.

Paragraph L97-102 lacks clarity on what was done exactly. The authors mentioned crop sampling points and planting. Please give precise details of what exactly was done in a precise and consistent manner. In other words, this section should provide readers with enough detail to replicate the study.

The hybrid name should be fully specified (L99). I suspect the full name is “BH540”!

L100 talks about traditional methods used for pest control. These need to be specified since they are not standard throughout the world.

Figures 2a and 2d are not referred to in the text.

L114 – statement which starts “Sun radiation…” needs to be rephrased.

L121, remove “can” and add “s” to the word “show”

L124, Table 1 remove “Explanation of the”

L128 replace “balancing” with “weighing”. Put unit of measurement after the word “plant” before converting to t/ha. Remove “gauging it with a balance”.

L129 Replace “conveying” with converting.

L131 Remove or rephrase “was articulated as a ton/ha”

L136 remove “values of”

Results and discussion

Most of the results were not discussed. Better to separate “Results” and “Discussions”. It would have helped and strengthen the manuscript if the authors do a combination of spectral bands on their own, VIs own their own and bands and VIs in prediction. See comment below (L218). For recent work on combining bands and/or VIs see: https://www.sciencedirect.com/science/article/pii/S0168169921001460

L148-150 which starts with “All soil …”. Where is this statement coming from and it is referenced?

L150-152 – Why quoting references when the authors are reporting their results? This can only be done when discussing! This also applies L175-177.

L156 Put an "=" sign between Min and minimum and the same applies to max and maximum (table legend)

L179-180 needs to be rephrased.

L194-195 remove “in the planting period”

L209 Start this statement as "The different relationships among SVIs and maize GY recorded (Fig 5 -6) could...."

L215 replace “bases” with “based”

L216-217 rephrase

L218 - Why did the authors not combine spectral bands and VIs and assess their predictive power rather than referring to literature? In their manuscripts, the authors used VIs as individual input variables to the model yet recent studies suggest that predictions can be improved when using a combination of bands or VIs in the model (see recent publication in Computers and Electronics in Agriculture journal): https://www.sciencedirect.com/science/article/pii/S0168169921001460

L218 – Please quote recent work – e.g.: https://www.sciencedirect.com/science/article/pii/S0168169921001460

L226 rephrase statement which begins “NDVI…”

L231 – give examples of “various factors”

L273 - The "red lines" inserted in the variable importance is not explained what it represents or serves. Otherwise, the two figures (r and s) are informative enough without the lines unless the authors want to show extra information depicted by the lines, which needs to be explained in the text or figure caption.

L289-290 Rephrase

L290-293 - This statement is not clear. How does predicting yield (a quantitative measure) has to do with time of harvesting? How do you locate any biotic and abiotic stresses through yield prediction? Rather you can be able to explain yield variation if you know the prevailing stresses and soil type.

L294 - Table 5 is not mentioned in the text. Also put a demarcating line before Mean.

For Use “LSD” instead of “MSD” and indicate at what p-value.

L299 replace “various” with “different”

L300 Add p-value for significant and not significantly immediately after the words “markedly” and “substantially”, respectively

L300 Put appropriate p-value. As it is, it means it was highly significant. It should be p > xx for example.

L306 what is the purpose of term “Tukey, p < 0.0001” here?

Conclusion

L314 replace “over with “in”

L315 Remove “novel” - these are not new VIs

L315 – statement beginning “However, ….” needs to be qualified by stating".... using our data" at the end of the statement

L318 The statement which begins “We obtained ..” is hanging.

L324-326 How? Not clear. Please explain clearly.

L326-328 Rephrase this recommendation. Studies on yield prediction have already been done at different phenological stages for certain stresses, e.g.: https://www.sciencedirect.com/science/article/pii/S0168169921001460

7. PLOS authors have the option to publish the peer review history of their article (what does this mean?). If published, this will include your full peer review and any attached files.

Reviewer #1: No

Reviewer #2: No

---

## [Author Response · Author response to Decision Letter 1]

24 May 2022

Date: June 01, 2022

Rebuttal letter

PONE-D-22-01589R1

We are happy about the academic editor and the reviewers’ comments, which strengthen the current version of the manuscript “A leaf reflectance-based crop yield modeling in Northwest Ethiopia”. In addition, our supreme sincere gratitude goes to you and the reviewers who devote their valuable time to bring our manuscript to a competent paper. 

We have provided a one by one reply to all concerns and comments given below. We thank you for your consideration of this resubmission and look forward to your response.

Best regards,

Gizachew Ayalew Tiruneh (on behalf of all co-authors)

Lecturer in Debre Tabor University

Ph.D. Fellow in soil science, Bahir Dar University

Email: tiruneh1972@gmail.com

 

Dear editor and reviewers, thank you so much for taking your valuable time to elevate the quality of our manuscript. We do hope that the editor’s and Reviewer’s concerns will be addressed.

Editor comments:

Comment 1: A rebuttal letter that responds to each point raised by the academic editor and reviewer(s). You should upload this letter as a separate file labeled 'Response to Reviewers'.

Response: We addressed the concerns provided by the editor and reviewers and uploaded a file labeled “Response to Reviewers”.

Comment 2: A marked-up copy of your manuscript that highlights changes made to the original version. You should upload this as a separate file labeled 'Revised Manuscript with Track Changes'.

Response: We tried to highlight our revised paper with tracked changes. We uploaded this as a separate file labeled 'Tracked changes'.

Comment 3: An unmarked version of your revised paper without tracked changes. You should upload this as a separate file labeled 'Manuscript'.

Response: We revised our manuscript without tracked changes. We uploaded this as a separate file labeled 'Manuscript'.

Comments 4: Response: We have not made any changes to financial disclosure.

Comments 5: If applicable, we recommend that you deposit your laboratory protocols in protocols.io to enhance the reproducibility of your results. Protocols.io assigns your protocol its own identifier (DOI) so that it can be cited independently in the future. For instructions see: https://journals.plos.org/plosone/s/submission-guidelines#loc-laboratory-protocols. Additionally, PLOS ONE offers an option for publishing peer-reviewed Lab Protocol articles, which describe protocols hosted on protocols.io. Read more information on sharing protocols at https://plos.org/protocols?utm_medium=editorial-email&utm_source=authorletters&utm_campaign=protocols

Response: Not applicable.

Comments 6: Please review your reference list to ensure that it is complete and correct. If you have cited papers that have been retracted, please include the rationale for doing so in the manuscript text, or remove these references and replace them with relevant current references. Any changes to the reference list should be mentioned in the rebuttal letter that accompanies your revised manuscript. If you need to cite a retracted article, indicate the article’s retracted status in the References list and also include a citation and full reference for the retraction notice.

Response: Thank you for your advice. We have checked that all references in the text are also in the reference and vice versa and all are complete and correct. We do not have retracted papers.

Reviewers' comments to Questions

Comments to the Author

Comment 1. If the authors have adequately addressed your comments raised in a previous round of review and you feel that this manuscript is now acceptable for publication, you may indicate that here to bypass the “Comments to the Author” section, enter your conflict of interest statement in the “Confidential to Editor” section, and submit your "Accept" recommendation.

Reviewer #1: All comments have been addressed

Reviewer #2: (No Response)

Response: Thank you the reviewer for your feedback. 

Comment 2. Is the manuscript technically sound, and do the data support the conclusions?

Reviewer #1: Yes

Reviewer #2: Yes

Response: Thank you. 

Comment 3: Has the statistical analysis been performed appropriately and rigorously?

Reviewer #1: Yes

Reviewer #2: No

Thank you. We have gone thoroughly the revised manuscript, and hopefully that the reviewers will be satisfied.

Comment 4. Have the authors made all data underlying the findings in their manuscript fully available?

Reviewer #1: Yes

Reviewer #2: Yes

Response: Thank you. 

Comment 5. Is the manuscript presented in an intelligible fashion and written in standard English?

Reviewer #1: Yes

Reviewer #2: No

Response: Thank you for your advice. We have thoroughly revised our manuscript with the help of Grammarly (premium) and licensed iThenticate software (as attached document), and we do hope that the reviewers concerns will be addressed.

Review Comments to the Author

Comments of Reviewer #1: Introduction

Page 3 line 45: Replace 'time taking' with 'time consuming'

Page 3 line 48 and 49: The two sentences on these lines begin with 'As a result...' consider revising one of the sentences.

Generally, the motivation in the introduction is still weak. The motivation should be strengthened.

Response: Thank you. We appreciate your valuable comments. We tried to address the comments and incorporated them in the revised manuscript in the introduction section. We hope that this revised version will be satisfying. 

Reviewer #2: Comments

Comment 1: Summary

The authors made partial corrections suggested in the first round of review. The manuscript still lacks details, mainly in the following areas: motivation/rationale in the introduction; English editing; clarity in materials and methods; further analysis using a combination of VIs in the model; referencing latest publications on yield modeling using remote sensing; and detailed discussion of results. Specific comments are given below. I recommend further revision.

Abstract

The last concluding statement (L35-37) talks about policy makers effectively managing resources! How?

Key words:

Why is “Arid” included as a key word when it is not mentioned anyway in the manuscript body except authors’ address and references? The study area receives rainfall ranging from 1,076 to 1,953 mm and cannot be classified as arid.

Response: Thank you for your valuable comments. We tried to address the comments and incorporated them in the revised manuscript in the abstract, introduction, materials and methods, combining of VIs in the model; referencing latest publications on yield modeling using remote sensing; and detailed discussion of results. Besides, we have thoroughly revised our manuscript with the help of Grammarly (premium) and licensed iThenticate software (as attached document), and we do hope that the reviewers concerns will be addressed.

Comment 2: Introduction

The introduction needs to be revised. The motivation/rationale is still weak or not convincing.

L49-51 should be taken to the second last paragraph of introduction and remove “As a result” from the statement.

L61-62 need rephrasing and qualifying.

L64 remove “as essential devices”

L75-76 remove “examined to”

L77, authors need to define the term “bund” for readers who are not familiar with the term/system

Response: Thank you for your comments. We tried to address the above comments and incorporated them in the revised manuscript including in the introduction section, and we do hope that the reviewers concerns will be addressed.

Comment 3. Material and Methods

The authors mentioned “midland”. This is not the right way to classify based on altitude. Better to say mid-altitude. “Midland” may mean a different thing altogether.

L86, the word “Teff” should not be in italics

The slope category is overlapping (L92-93): gentle (2–5 %), sloping (5–10 %), and strongly sloping (10–15 %). Please correct. I also suggest the mid slope (5-10%) category to be referred to as “medium” sloping rather than just “sloping” with corrections on overlapping categories.

Paragraph L97-102 lacks clarity on what was done exactly. The authors mentioned crop sampling points and planting. Please give precise details of what exactly was done in a precise and consistent manner. In other words, this section should provide readers with enough detail to replicate the study.

The hybrid name should be fully specified (L99). I suspect the full name is “BH540”!

L100 talks about traditional methods used for pest control. These need to be specified since they are not standard throughout the world.

Figures 2a and 2d are not referred to in the text.

L114 – statement which starts “Sun radiation…” needs to be rephrased.

L121, remove “can” and add “s” to the word “show”

L124, Table 1 remove “Explanation of the”

L128 replace “balancing” with “weighing”. Put unit of measurement after the word “plant” before converting to t/ha. Remove “gauging it with a balance”.

L129 Replace “conveying” with converting.

L131 Remove or rephrase “was articulated as a ton/ha”

L136 remove “values of”

Response: Thank you for your suggestion. We tried to address the above comments and incorporated them in the revised manuscript including in the Material and Methods section, and we do hope that the reviewers concerns will be addressed.

Comment 4. Results and discussion

Most of the results were not discussed. Better to separate “Results” and “Discussions”. It would have helped and strengthen the manuscript if the authors do a combination of spectral bands on their own, VIs own their own and bands and VIs in prediction. See comment below (L218). For recent work on combining bands and/or VIs see: https://www.sciencedirect.com/science/article/pii/S0168169921001460

L148-150 which starts with “All soil …”. Where is this statement coming from and it is referenced?

L150-152 – Why quoting references when the authors are reporting their results? This can only be done when discussing! This also applies L175-177.

L156 Put an "=" sign between Min and minimum and the same applies to max and maximum (table legend)

L179-180 needs to be rephrased.

L194-195 remove “in the planting period”

L209 Start this statement as "The different relationships among SVIs and maize GY recorded (Fig 5 -6) could...."

L215 replace “bases” with “based”

L216-217 rephrase

L218 - Why did the authors not combine spectral bands and VIs and assess their predictive power rather than referring to literature? In their manuscripts, the authors used VIs as individual input variables to the model yet recent studies suggest that predictions can be improved when using a combination of bands or VIs in the model (see recent publication in Computers and Electronics in Agriculture journal): https://www.sciencedirect.com/science/article/pii/S0168169921001460

L218 – Please quote recent work – e.g.: https://www.sciencedirect.com/science/article/pii/S0168169921001460

L226 rephrase statement which begins “NDVI…”

L231 – give examples of “various factors”

L273 - The "red lines" inserted in the variable importance is not explained what it represents or serves. Otherwise, the two figures (r and s) are informative enough without the lines unless the authors want to show extra information depicted by the lines, which needs to be explained in the text or figure caption.

L289-290 Rephrase

L290-293 - This statement is not clear. How does predicting yield (a quantitative measure) has to do with time of harvesting? How do you locate any biotic and abiotic stresses through yield prediction? Rather you can be able to explain yield variation if you know the prevailing stresses and soil type.

L294 - Table 5 is not mentioned in the text. Also put a demarcating line before Mean.

For Use “LSD” instead of “MSD” and indicate at what p-value.

L299 replace “various” with “different”

L300 Add p-value for significant and not significantly immediately after the words “markedly” and “substantially”, respectively

L300 Put appropriate p-value. As it is, it means it was highly significant. It should be p > xx for example.

L306 what is the purpose of term “Tukey, p < 0.0001” here?

Response: Thank you for your suggested reference. We tried to address the above comments and incorporated them in the revised manuscript including in the Results and discussion section, and we do hope that the reviewers concerns will be addressed.

Comment 5. Conclusion

L314 replace “over with “in”

L315 Remove “novel” - these are not new VIs

L315 – statement beginning “However, ….” needs to be qualified by stating".... using our data" at the end of the statement

L318 The statement which begins “We obtained ..” is hanging.

L324-326 How? Not clear. Please explain clearly.

L326-328 Rephrase this recommendation. Studies on yield prediction have already been done at different phenological stages for certain stresses, e.g.: https://www.sciencedirect.com/science/article/pii/S0168169921001460

Response: Thank you for your comments and suggested reference. We tried to address the above comments and incorporated them in the revised manuscript including in the Conclusion section, and we do hope that the reviewers concerns will be addressed.

Comment 6. PLOS authors have the option to publish the peer review history of their article (what does this mean?). If published, this will include your full peer review and any attached files.

Do you want your identity to be public for this peer review? For information about this choice, including consent withdrawal, please see our Privacy Policy.

Reviewer #1: No

Reviewer #2: No

Response: Thank you. We have used PACE with this submission, so this should be right.

Please note that once again, thank you very much. Your comments are greatly appreciated.

Best regards,

Gizachew Ayalew Tiruneh (on behalf of all co-authors)

Lecturer in Debre Tabor University

Ph.D. Fellow in soil science, Bahir Dar University

Email: tiruneh1972@gmail.com

---

## [Editor Report · Decision Letter 2]

31 May 2022

A leaf reflectance-based crop yield modeling in Northwest Ethiopia

PONE-D-22-01589R2

Dear Dr. Tiruneh,

We’re pleased to inform you that your manuscript has been judged scientifically suitable for publication and will be formally accepted for publication once it meets all outstanding technical requirements.

Kind regards,

Abel Chemura

Academic Editor

PLOS ONE
---

## [Editor Report · Acceptance letter]

3 Jun 2022

PONE-D-22-01589R2 

A leaf reflectance-based crop yield modeling in Northwest Ethiopia 

Dear Dr. Tiruneh:

I'm pleased to inform you that your manuscript has been deemed suitable for publication in PLOS ONE. Congratulations! Your manuscript is now with our production department. 

Kind regards, 

on behalf of

Dr. Abel Chemura 

Academic Editor

PLOS ONE